

# Oomycete metabarcoding reveals the presence of *Lagenidium* spp. in phytotelmata

Paula Leoro-Garzon, Andrew J. Gonedes, Isabel E. Olivera and Aurélien Tartar

Department of Biological Sciences, Nova Southeastern University, Fort Lauderdale, FL,
United States of America

## ABSTRACT

The oomycete genus *Lagenidium*, which includes the mosquito biocontrol agent *L. giganteum*, is composed of animal pathogens, yet is phylogenetically closely related to the well characterized plant pathogens *Phytophthora* and *Pythium* spp. These phylogenetic affinities were further supported by the identification of canonical oomycete effectors in the *L. giganteum* transcriptome. In this study, culture-independent, metabarcoding analyses aimed at detecting *L. giganteum* in bromeliad phytotelmata (a proven mosquito breeding ground) microbiomes were performed. Two independent and complementary microbial detection strategies based on the amplification of *cox1* DNA barcodes were used and produced globally concordant outcomes revealing that two distinct *Lagenidium* phylotypes are present in phytotelmata. A total of 23,869 high quality reads were generated from four phytotelmata, with 52%, and 11.5% of these reads taxonomically associated to oomycetes, and *Lagenidium* spp., respectively. Newly designed *Lagenidium*-specific *cox1* primers combined with cloning/Sanger sequencing produced only *Lagenidium* spp. sequences, with a majority of variants clustering with *L. giganteum*. High throughput sequencing based on a Single Molecule Real Time (SMRT) approach combined with broad range *cox1* oomycete primers confirmed the presence of *L. giganteum* in phytotelmata, but indicated that a potentially novel *Lagenidium* phylotype (closely related to *L. humanum*) may represent one of the most prevalent oomycetes in these environments (along with *Pythium* spp.). Phylogenetic analyses demonstrated that all detected *Lagenidium* phylotype *cox1* sequences clustered in a strongly supported, monophyletic clade that included both *L. giganteum* and *L. humanum*. Therefore, *Lagenidium* spp. are present in phytotelmata microbiomes. This observation provides a basis to investigate potential relationships between *Lagenidium* spp. and phytotelma-forming plants, and reveals phytotelmata as sources for the identification of novel *Lagenidium* isolates with potential as biocontrol agents against vector mosquitoes.

## INTRODUCTION

Oomycetes are heterotrophic eukaryotes that are morphologically similar to fungi but phylogenetically related to diatoms and brown algae, and grouped with these photosynthetic relatives within the phylum Heterokonta (*Derevnina et al., 2016*; *Kamoun et al., 2015*). The

Corresponding author
Aurélien Tartar, aurelien@nova.edu

best-characterized oomycetes are disease-causing agents with significant impacts on human activities and food security, and the majority of the work directed at understanding the biology of oomycetes is aimed at controlling or eliminating these organisms from anthropogenic agroecosystems such as crop fields or aquaculture facilities (*Derevnina et al., 2016*). A minority of oomycetes have potential as biological control agents, including the mycoparasite *Pythium oligandrum* (*Horner, Grenville-Briggs & Van West, 2012*) and the mosquito pathogen *Lagenidium giganteum* (*Kerwin, Dritz & Washino, 1994*), and have been developed as the commercial products Polyversum and Laginex, respectively. However, safety concerns over the true host range of *L. giganteum* (*Vilela et al., 2015*) have prompted a shift from large-scale production and commercialization to molecular explorations directed at identifying bioactive compounds that may be translated into novel mosquito control strategies (*Singh & Prakash, 2010*). The recent transcriptome analyses of *L. giganteum* have also contributed in expanding the characterization of oomycete diversity at the molecular level (*Olivera et al., 2016*; *Quiroz Velasquez et al., 2014*). Sequence analyses suggested that *L. giganteum* evolved from plant pathogenic ancestors and has retained genes typically associated with plant tissues infections, such as the CRN or CBEL effectors that have been extensively characterized in *Phytophthora infestans* and related plant pathogenic species. In addition, the *L. giganteum* transcriptome was shown to contain several genes that were absent from plant pathogenic genomes, and that were conserved either in entomopathogenic eukaryotes (*Quiroz Velasquez et al., 2014*), or in animal pathogenic oomycetes (*Olivera et al., 2016*).

The emerging dichotomy reflected by the *L. giganteum* transcriptome is reminiscent of the most recent analyses of fungal entomopathogens genomes, and suggests that similarities between fungal and oomycetes entomopathogens may be extended from morphology and pathological strategies to evolutionary history and ecological relationships. Genomic analyses have demonstrated that two of the most common genera of insect-pathogenic fungi, *Metarhizium* and *Beauveria*, have evolved from plant pathogens, and have retained genes indicative of plant interactions (*Moonjely, Barelli & Bidochka, 2016*; *Wang, Leger & Wang, 2016*). In fact, both *Metarhizium* and *Beauveria* spp. are now widely regarded as plant endophytes that maintain significant symbiotic relationships with their plant hosts, where insect infections, and subsequent nitrogen transfer from insect to plant tissues (*Behie & Bidochka, 2014*), may play only a small role among the diverse beneficial interactions that have been shown to result from the presence of these fungi in plants and their rhizospheres (*Lopez & Sword, 2015*; *Sasan & Bidochka, 2012*). In agreement with these recent studies, the oomycete *L. giganteum* have been hypothesized as a potential endophyte that can alternate between plant and insect hosts, and has the genomic resources to engage in both type of relationships (*Quiroz Velasquez et al., 2014*). Most *Lagenidium* spp. isolations have followed episodic observations of colonization in various animal host tissues (*Mendoza et al., 2016*; *Nakamura, Nakamura & Hatai, 1995*; *Vilela et al., 2019*), and therefore, to date, there is little evidence of meaningful ecological associations between *Lagenidium* spp. and plants. However, phytotelmata (small bodies of water impounded by plants) appear as likely habitats for *Lagenidium* spp, based on a previous study that

reported *Lagenidium*-infected invertebrates in plant axils (*Frances, Sweeney & Humber, 1989*), and on the well-established knowledge that phytotelmata represent ideal breeding grounds for *L. giganteum* potential hosts, including mosquitoes (*Derraik, 2009*). The role of phytotelmata as mosquito breeding sites has been recently highlighted by South Florida-based studies indicating that *Aedes aegypti* mosquitoes (the main vectors for dengue fever, yellow fever and zika) may successfully evade vector control strategies by breeding in popular and difficult-to-treat ornamental bromeliads (*Wilke et al., 2018*).

To test the hypothesis that *Lagenidium giganteum* inhabit phytotelmata (especially, South Florida bromeliad phytotelmata) and therefore may establish tripartite interactions with both insect and plant hosts, a culture-independent assay aimed at detecting *Lagenidium* spp. sequences (metabarcoding) was developed. Molecular-based approaches based on the PCR amplification of selected gene fragments have been used for multiple phyla and multiple environments (*Abdelfattah et al., 2018*), and a wealth of information have been compiled in databases such as the Barcode Of Life Data system (*Ratnasingham & Hebert, 2007*). Standard barcodes consist of *cox1* and ITS gene regions for animals and fungi, respectively, whereas plant barcoding has relied on multiple chloroplastic markers (*Adamowicz, 2015*). A barcode consensus for oomycetes has yet to emerge. Previous studies have proposed and tested several potential candidate genes, including the ITS region (*Gómez et al., 2019*; *Riit et al., 2016*; *Robideau et al., 2011b*), and the *cox1*, *cox2*, and cytochrome *b* genes (*Choi et al., 2015*; *Giresse et al., 2010*; *Robideau et al., 2011b*). Most of these oomycete barcoding efforts have been restricted to assessing phylum-specific primers on DNA preparations obtained from axenically-grown isolates, and few have transitioned to primer validation assays that (i) incorporated environmental sampling, and (ii) combined primers with specific sequencing strategies/platforms. Pioneer oomycete metabarcoding studies have favored the use of ITS primers, and the production of small size amplicons (*Prigigallo et al., 2016*; *Riit et al., 2016*; *Sapkota & Nicolaisen, 2015*). Oomycete metagenomics has yet to fully integrate third generation sequencing technologies that enable long read analyses, despite recent studies demonstrating that strategies such as the Single Molecule Real Time (SMRT) method developed by Pacific Biosciences (known as PacBio sequencing) delivered similar barcoding sequencing performances compared to other platforms while producing much longer (and therefore more informative) DNA barcodes (*Pootakham et al., 2017*; *Wagner et al., 2016*). These improvements in long read sequencing quality provide a renewed opportunity to assess the *cox1* gene as a oomycete barcode, since oomycete-specific *cox1* primers have already been published, and they produce the longest (>600bp) oomycete barcode evaluated to date (*Choi et al., 2015*). In light of this new possibility, the purpose of this study was two-fold: first, to develop *Lagenidium giganteum*-specific *cox1* primers to assess the presence of this entomopathogenic oomycete in bromeliad phytotelmata, and second, to couple the use of previously published oomycete-specific *cox1* primers with SMRT-based sequencing strategy, and assess the potential of this combination to not only confirm the presence of *L. giganteum* in phytotelmata, but also evaluate the relative abundance of *L. giganteum* among other phytotelmata-inhabiting oomycete species.

## MATERIALS AND METHODS

### Oomycete cultures, *cox1* gene sequencing, and genus-specific primer design

The *Lagenidium giganteum* strain ARSEF 373 was accessed from the USDA Agricultural Research Service Collection of Entomopathogenic Fungal Cultures (ARSEF, Ithaca, NY) and was grown in a defined Peptone-Yeast-Glucose (PYG) media supplemented with 2mM $CaCl_2$, 2mM $MgCl_2$ and 1ml/L soybean oil (*Kerwin & Petersen, 1997*). Axenic cultures were processed for genomic DNA extraction using the Qiagen DNeasy minikit, as previously described (*Olivera et al., 2016*; *Quiroz Velasquez et al., 2014*). The genomic DNA preparations were used as templates in Polymerase Chain Reactions (PCR) in combination with the oomycete-specific *cox1* primers OomCoxI-Levup (5′-TCAWCWMGATGGCTTTTTTCAAC-3′) and OomCoxI-Levlo (5′-CYTCHGGRTGWCCRAAAAACCAAA-3′). These primers were designed to overlap the standard *cox1* DNA barcode used in other groups and recommended by the Consortium for the Barcode of Life (CBOL) initiative (*Robideau et al., 2011b*). PCR conditions corresponded to the following pattern repeated for 30 cycles: 95 °C for 30 s, 50 °C for 30 s, and 72 °C for 1 min. The resulting products were purified using the QIAquick PCR purification Kit (Qiagen, USA) and sequenced commercially using traditional Sanger technology (Macrogen USA). The generated sequences were aligned with homologous oomycete sequences obtained from the Barcode of Life Data System (BOLD) database of *cox1* genes (*Ratnasingham & Hebert, 2007*). Alignments were performed using ClustalX with default parameters (*Larkin et al., 2007*). The *cox1* gene alignment was used to visually identify regions suitable for genus- or species-specific primer design. Alignments corresponding to selected locations were used as inputs for the construction of sequence logos using WebLogo, version 3 (*Crooks et al., 2004*).

### Phytotelmata sampling and plant identification

Phytotelmata were sampled from ornamental plants on the Nova Southeastern University (NSU) main campus in Fort Lauderdale, FL, USA. Four plants were selected based on two criteria, including a visual, tentative taxonomic characterization of plants as bromeliads, and the observable presence of a large volume of water within the plants axils. The precise location of each plant was recorded using the Global Position System (GPS). Phytotelmata samples consisted of a 100 mL volume of water collected using sterile serological pipettes, and transferred in sterile 50 mL conical tubes. The water samples were inspected visually for the presence of macroscopic debris and invertebrates. In addition, leaf tissues (2 to 3 cm$^2$) were also sampled for each plant, in an effort to associate phytotelmata samples with plant taxonomic classification. The leaf samples were grounded in liquid nitrogen and processed for DNA extraction using the Qiagen DNeasy Plant Mini kit (according to the manufacturer's instructions). The plant genomic DNA preparations were used to PCR-amplify plant barcodes using primers designed for previously characterized loci, including the *trnH*-*psbA* spacer region (*Kress & Erickson, 2007*; *Kress et al., 2005*) and the internal transcribed spacer (ITS) region of nuclear rDNA (*Cheng et al., 2016*) traditionally

used for a wide variety of land plants, as well as the *trnC-petN* spacer marker used more specifically for bromeliad barcoding (*Versieux et al., 2012*).

## Phytotelmata microbiomes DNA extractions and *cox1* barcode amplification

Phytotelmata samples were vacuum-filtered through 47 mm diameter, 0.45 μm pore size nitrocellulose filters (Millipore), as previously described (*Mancera et al., 2012*), and the microbial fauna retained on these filters was subjected to DNA extraction using the MoBio PowerWater DNA isolation kit (according to the manufacturer's instructions). A similar workflow (vacuum filtration and DNA extraction) was used to process negative control water samples. These samples consisted of 100 mL of water collected at a drinking water fountain located on the NSU campus, as well as a 100 mL of seawater collected off the coast of Hollywood Beach, FL, USA. The resulting metagenomic DNA preparations obtained from phytotelmata and negative controls samples were initially PCR amplified using the oomycete-specific *cox1* primers OomCoxI-Levup and OomCoxI-Levlo and the reaction parameters described above. Products of these PCR reactions were visualized on agarose gels. Subsequently, aliquots (one μl, non purified) corresponding to the products obtained using the OomCoxI-Lev primer set were used as templates for a second round of amplification. These nested PCR reactions were performed using the *Lagenidium*-specific primers (LagCox F & R) under stringent conditions (30 cycles of the following pattern: 95 °C for 30 s, 68 °C for 30 s, and 72 °C for 1 min). Products of these PCR reactions (expected size: 525 bp, nested within the 700-bp products obtained using the OomCoxI-Lev set) were visualized on agarose gels, cloned using the Invitrogen TOPO technology and processed for commercial Sanger sequencing (Macrogen USA). Resulting sequences were evaluated through homology searches and phylogenetic analyses as described below.

## Oomycete community assessment through *cox1* metabarcoding

The phytotelmata *cox1* libraries were prepared for single molecule real time (SMRT) sequencing using recommended protocols available from Pacific Biosciences (PacBio multiplexed SMRTbell libraries). The workflow included a two-step PCR amplification as previously published (*Pootakham et al., 2017*). First, fusion primers were custom designed by combining the OomCoxI-Levup and OomCoxI-Levlo primer sequences described above with the PacBio universal sequence. These primers were HPLC purified and further modified by the addition of a 5′ block (5′-NH4, C6) to ensure that carry-over amplicons from the first round of PCR were not ligated in the final libraries (Integrated DNA Technologies). The first PCR reaction used these primers to amplify *cox1* fragments from all four phytotelmata metagenomic DNA preparations. Resulting products were gel-extracted and served as templates for the second PCR reactions. The second reaction used the PacBio Barcoded Universal Primers (BUP) so that unique combinations of (symmetrical) forward and reverse barcoded primers were associated with each phytotelmata samples. Products of the second amplification were purified (DCC, Zymo Research), and sent to the University of Florida Interdisciplinary Core for Biotechnology Research (ICBR) where amplicons were pooled in equimolar concentrations and further processed for library construction and

SMRT sequencing. The PacBio raw reads were demultiplexed and assessed for quality at the ICBR. Quality control processing included eliminating poor quality sequences, sequences outside the expected amplification size (ca. 810 bp) and sequences that failed to include both flanking, symmetrical barcodes. High quality reads served as inputs for homology searches to assign taxonomic identification down to the genus level, using BLAST2GO (*Conesa et al., 2005*). Sequences homologous to *Lagenidium* spp. were further processed for thorough phylogenetic analyses. These sequences were trimmed to eliminate flanking 5′ and 3′ regions, and evaluated for redundancy (100% homology) and OTU clustering using the ElimDupes tool (http://www.hiv.lanl.gov/). Selected sequences were included in the alignment described below.

## Phylogenetic analyses

The *cox1* gene sequences generated from axenic cultures and environmental samples were aligned with homologous oomycete sequences using ClustalX (*Larkin et al., 2007*). Most orthologous sequences were downloaded from the BOLD database (*Ratnasingham & Hebert, 2007*) as described above. However, the alignment was also complemented with orthologous *Lagenidium* spp. sequences available from GenBank, including the *cox1* sequenced fragments recently generated from *Lagenidium* spp. isolates collected on mammalian tissues (*Spies et al., 2016*). The complete *cox1* alignment consisted of a 620-character dataset that contained 62 taxa. The position of the shorter, Sanger-based environmental sequences was inspected visually and confirmed based on the location of the *Lagenidium*-specific primers. The jModeltest program (*Darriba et al., 2012*) was used to identify the most appropriate maximum likelihood (ML) base substitution model for this dataset. The best-fit model consistently identified by all analyses was the Generalized Time Reversible model with a gamma distribution for variable sites, and an inferred proportion of invariants sites (GTR+G+I). ML analyses that incorporated the model and parameters calculated by jModeltest were performed using PhyML3.0 (*Guindon et al., 2010*). ML bootstrap analyses were conducted using the same model and parameters in 1,000 replicates. The phylogenetic tree corresponding to the ML analyses was edited using FigTree v. 1.4.4.

## RESULTS

### *Lagenidium giganteum cox1* gene sequence analysis

The cox1 fragment generated from the *Lagenidium giganteum* strain ARSEF373 was 683 bp long, and its sequence was deposited in the GenBank/EMBL/DDBJ databases under the accession number MN099105. Homology searches (not shown) demonstrated that the generated sequence was 100% identical to *cox1* sequences reported from two other strains of *L. giganteum* (strains ATCC 52675, and CBS 58084, with *cox1* sequences publicly accessible under the accession numbers KF923742 and HQ708210, respectively). Both strains ARSEF 373 and ATCC 52675 were originally isolated from mosquito larvae, according to culture collection records. Further comparisons (not shown) indicated that sequences from these mosquito-originating strains appeared divergent from the *cox1* fragments sequences generated from multiple strains of *L. giganteum* f. *caninum* that have been reported as

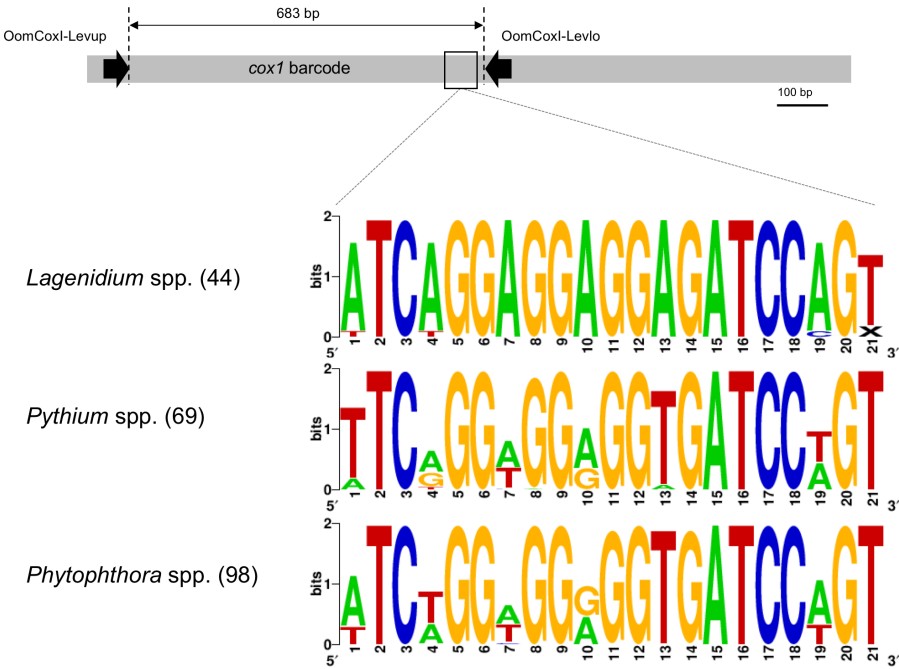

**Figure 1** **Schematic representation of the *cox1* gene as a metabarcoding target.** Previously developed, oomycete-specific primers, named OomCoxI-LevUp and OomCoxI-LevLo, were designed to amplify the 5′ end portion of the gene that is typically used as barcode (sometimes referred to as the ''Folmer region'', especially in metazoans). Oomycete *cox1* sequences obtained using these primers were aligned and evaluated for sites compatible with the development of *Lagenidium* genus-specific primers. As illustrated by the sequence logos, a locus immediately upstream of the OomCox1-LevLo location showed genus-level specificity and was selected for primer design. The logos correspond to the complete primer location (20 bp). Numbers in parentheses indicate the total number of sequences (for each genus) used to generate the logos.

mammal pathogens, yet also retained the ability to infect mosquito in laboratory settings (*Vilela et al., 2015*). These results highlight the potential of molecular barcodes such as *cox1* to distinguish between the known *Lagenidium* strains.

Unsurprisingly, the entomopathogenic *L. giganteum* cox1 sequences were also different from sequences characterizing more phylogenetically-distant oomycetes, including *Lagenidium*, *Pythium* and *Phytophthora* spp., as well as other Peronosporales. These differences provided a basis to develop *Lagenidium giganteum*-specific primers, and the location ultimately selected for primer design is illustrated in Fig. 1. The specificity of the designed primers relied especially on the reverse primer, that is located on a region that is immediately (40 bp) upstream the OomCoxI-Levlo primer (Fig. 1). This region was characterized by the presence of a 5′-ATCA-3′ motif that was showed to be prevalent in *Lagenidium*: alignments demonstrated that it was present on all the publicly available *cox1* sequences (41 sequences total) obtained from *L. giganteum* (both mosquito and mammal strains) as well as *L. humanum* (Fig. 1). In contrast, the motif was not found in *L. deciduum* sequences (3 sequences), and was found only sporadically in *Pythium* and *Phytophthora* sequences (most notably in *Py. helicandrum*,

*Py. carolinianum*, and some strains of *P. ramorum*, *P. cactorum* and *P. infestans*). As a result, the reverse *Lagenidium*-specific primer was designed to incorporate the reverse complement sequence 5′-TGAT-3′ at its 3′ end, and overlapped additional polymorphic sequences between *Lagenidium* and other Peronosporales. The primer sequences were finalized at 5′-ACTGGATCTCCTCCTCCTGAT-3′ for the reverse primer (LagCox R), and 5′-TAACGTGGTTGTAACTGCAC-3′ for the matching forward primer (LagCox F).

## Environmental detection of *Lagenidium* spp. in phytotelmata using Sanger sequencing

A total of four plants were selected for analysis (Fig. 2), and information about these plants taxonomic identification through barcoding is available in File S1. The oomycete- and *Lagenidium*-specific *cox1* primers were used in combination with metagenomic DNA preparations representative of the four plant phytotelmata (Fig. 2E). As illustrated in Fig. 2, the first round of amplification, using oomycete- specific cox1 primers, consistently produced detectable amplicons of the expected size (ca. 700 bp) for all plant-based water sources, but not the control water sources, suggesting the presence of oomycetes in the four sampled phytotelmata. Similarly, the nested PCR amplifications, using *Lagenidium*-specific primers (Fig. 1) and stringent PCR conditions, also produced fragments of the expected, 525 bp- size, leading to the production of twelve high-quality sequences (three per plants) publicly available in GenBank under the accession numbers MN099114–MN099125. Homology searches demonstrated that all twelve of these newly-obtained, environmental sequences were more similar to *Lagenidium* spp. *cox1* sequences than other any oomycete barcodes (not shown). However, alignments also revealed that none of the environmental sequences were 100% identical to the previously published *Lagenidium* spp. *cox1* sequences obtained from known strains maintained in axenic cultures (based on the 484 bp fragment length), suggesting a yet-unsampled diversity within the *Lagenidium* genus. Using a traditional 97% distance level to build Operational Taxonomic Unit (OTUs), the twelve Sanger-based sequences clustered in two distinct OTUs. The first OTU consisted of the *Lagenidium humanum cox* 1 barcode (accession number KC741445) clustered with the three sequences obtained from P3 (these three sequences were identical) and two identical sequences from the P1 phytotelma. All other environmental sequences (three identical sequences from the P4 phytotelma, as well as one unique sequence from P1, and three unique sequences from P2) clustered in a second OTU that included all known cox1 sequences from *L. giganteum*, including the *L. giganteum* f. *caninum* cox1 barcodes. These preliminary findings strongly suggested that all environmental sequences corresponded to *Lagenidium* spp. *cox1* genes, and that the mosquito pathogen *Lagenidium giganteum* is present in phytotelmata. Relative abundance calculations indicated that *L. giganteum* (7 reads out of 12, or 58%) appears more frequently than *L. humanum*-like isolates (5 reads out of 12, or 42%). The sampled sequences, albeit limited in number, also validated the newly designed primers as specific for the genus *Lagenidium*. All sequences were incorporated in the phylogenetic analyses described below, in an effort to more precisely determine their taxonomic nature.

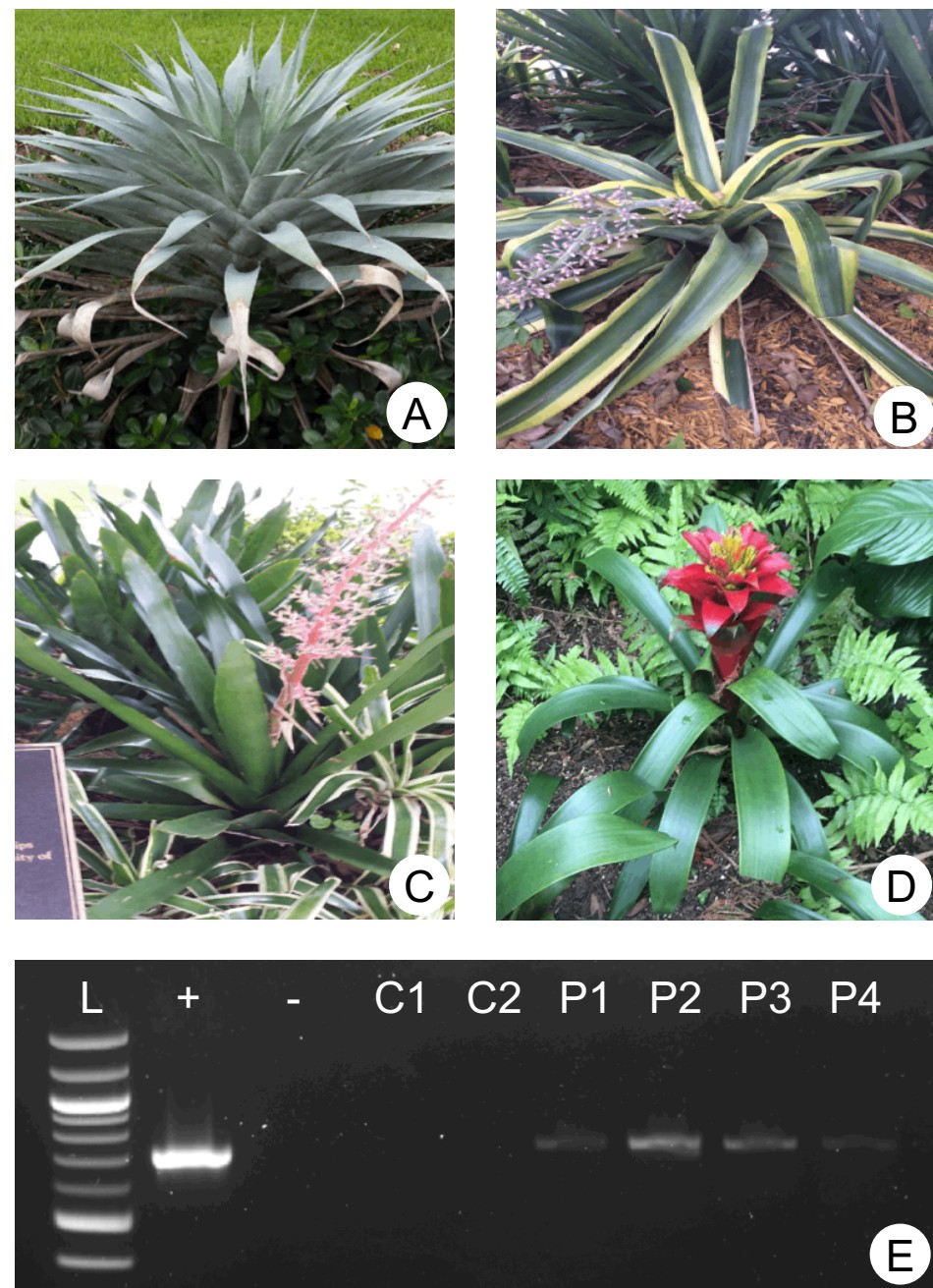

**Figure 2 Sampled plants and molecular detection of phytotelmata oomycetes.** (A–D) depict the four plants (used as ornamentals on the NSU campus) representing the origin of the phytotelmata samples denoted P1 to P4 throughout the study (plants A-D = phytotelmata P1–P4, respectively). Environmental DNA was extracted from these four plant phytotelmata and tested for the presence of oomycetes using cox1 primers. (E) illustrates PCR products generated using these environmental DNA preparations as templates combined with the oomycete-specific cox1 primers (OomCoxI-LevUp and OomCoxI-LevLo). Phytotelmata metagenomic DNA preparations are labelled as P1–P4, while (+) and (−) lanes represent positive (*L. giganteum* DNA) and negative (no template) control. Additional control reactions (C1, C2) included templates corresponding to metagenomic DNA extracted from water fountain (tap) and ocean waters, respectively. Visible PCR products for lanes P1–P4 demonstrated that oomycetes were readily detected in all sampled phytotelmata.

## Assessment of *Lagenidium* spp. presence in phytotelmata microbiome using *cox1* PacBio sequencing

A total of 23,857 reads were retained, demultiplexed and processed for bioinformatics analyses (File S1. Analyzed PacBio sequence datasets (available in the NCBI Sequence Read Archive data under accession numbers SRX6359420–SRX6359423 as part of Bioproject PRJNA550619) included 7,852, 6,576, 5,151 and 4,278 reads for phytotelmata P1 to P4, respectively. Homology searches indicated that only a minority of these filtered reads (227 reads, or 0.9%) could not be assigned a taxonomic classification at the phylum/genus levels. Most sequences were classified into two major eukaryotic phyla, corresponding to animals and protists (Fig. 3). Animal sequences appeared to exclusively belong to insects and related taxa (Fig. 3), consistent with the hypothesis that phytotelmata are actively used environments for a specialized fauna of invertebrates. Protist sequences were further divided into oomycete and non-oomycete subgroups, and, as anticipated, oomycete sequences represented the majority of protist sequences in three sampled communities (Fig. 3). Oomycetes were found especially prevalent in phytotelmata P3 and P4, where they accounted for 79 and 90% of the sequences, respectively. Oomycetes represented 49% of the sequences in the P1 phytotelma, where the sequence distribution was characterized by a large proportion (40%) of invertebrate sequences (Fig. 3). These invertebrate sequences virtually all corresponded to a single OTU closely related to an unidentified Arachnida *cox1* barcode (data not shown). In contrast to the P1, P3 and P4 samples, the P2 filtered reads contained a majority of non-oomycete sequences (Fig. 3), with an overrepresentation (82%) of OTUs homologous to the freshwater diatom genus *Sellaphora* (not shown). Oomycete sequences in P2 represented only 12% of the total sequences generated for this phytotelma (Fig. 3). These results pointed to the promises of using SMRT-based, long read *cox1* sequences to assess the oomycete communities of selected environments but also suggested that the primer sequences, or the amplification conditions, used for these analyses may need to be refined in order to limit the production of amplicons from organisms that are phylogenetically close to oomycetes, such as diatoms. Overall, oomycete barcodes were detected in all phytotelmata, and sequence classifications at the genus level revealed a total of 10 oomycete genera, including *Achlya*, *Aphanomyces*, *Halophytophthora*, *Haptoglossa*, *Lagenidium*, *Phytophthora*, *Phytopythium*, *Pythiogeton*, *Pythium* and *Saprolegnia*. As illustrated in Fig. 3, *Pythium*, followed by *Lagenidium*, represented the most prevalent genera in the oomycete communities of all phytotelmata. Sequences identified as *Pythium* spp. corresponded to 90%, 93%, 39% and 97% of all oomycetes reads for plants P1–P4, respectively (Fig. 3) In agreement with the Sanger-based analyses, sequences homologous to *Lagenidium* spp. *cox1* barcodes were detected in all samples. These sequences accounted for 7.2%, 1.7%, 59.8% and 0.3% of all oomycete reads, for phytotelmata P1 to P4, respectively, indicating that *Lagenidium* was present at low frequencies when compared to *Pythium*, except in the case of the P3 sample (Fig. 3). Also in agreement with the Sanger-based analyses, none of the reads identified as *Lagenidium* spp. were identical to the previously published *L. humanum* cox1 sequence fragment. However, a small number of reads (<1%) were shown to be 100% homologous to the mosquito pathogen *L. giganteum* cox 1 gene sequence (accession numbers HQ708210 and KF923742): 3 reads (out of 279) in the P1

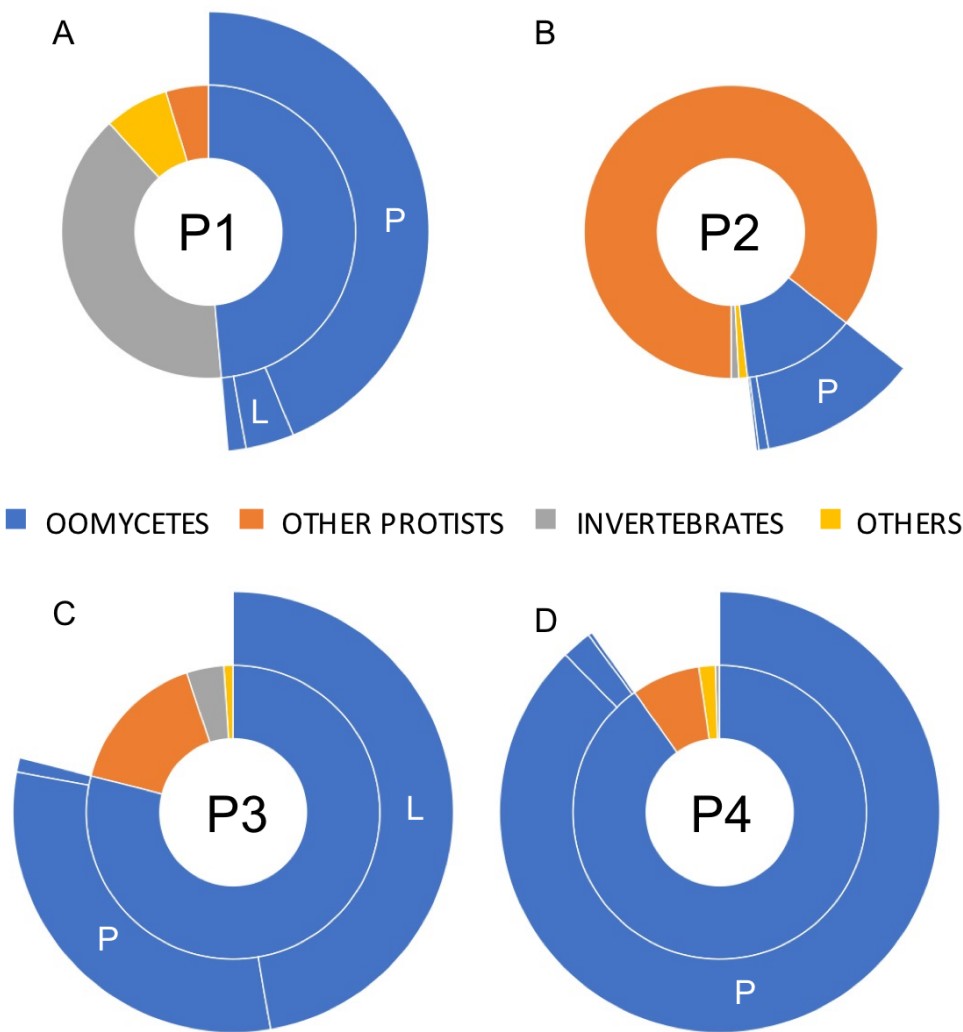

**Figure 3  Relative taxonomic distribution of *cox1* sequences generated using the PacBio sequencing technology platform.** The four sampled phytotelmata are denoted as P1–P4 in the circle centers. As anticipated, the majority of sequences showed similarities to oomycete DNA barcodes (color coded in blue), although sequences corresponding to non-target taxonomic groups were also detected. For oomycetes, a genus-level taxonomic break-down (outer circle portions) demonstrated that the most prevalent genera in phytotelmata were *Pythium* and *Lagenidium*, represented by letters P and L, respectively. All other oomycetes were regrouped into the third classification (i.e., not P nor L). For clarity purposes, letters corresponding to oomycete genera are not indicated when the overall distribution frequency is below 5%.

sample and 1 read (out of 2,345) in the P3 dataset. OTU clustering at 100% distance level recognized identical reads within and between samples, and revealed that a single sequence was consistently the most predominant *Lagenidium* barcode across all four phytotelmata: this predominant sequence was represented by 103 reads out of 279 (37%) for P1, 3 reads out of 14 (21%) for P2, 1,215 reads out of 2,435 (50%) for P3 and 3 reads out of 13 (23%) for P4. Using a lower distance level for OTU clustering (97%), virtually all PacBio reads (>99%) clustered with these predominant sequences (not shown), and were associated with

the *L. humanum* barcode. Finally, further sequence alignments compared reads obtained through Sanger vs. PacBio technologies. These comparative analyses showed that the overrepresented PacBio reads for P1-P4 were 100% identical to the sequences obtained using Sanger-based technologies for the P3 sample, highlighting the concordance between the two *Lagenidium* spp. barcode detections.

## Phylogenetic analyses

The generation of novel *Lagenidium*-like *cox1* sequences using both traditional and Next-Generation sequencing technologies prompted comprehensive phylogenetic analyses that incorporated these environmental barcodes within a robust alignment of sequences obtained from axenic cultures. The phylogram inferred from Maximum Likelihood analyses (ML) is presented in Fig. 4. The tree was rooted with representatives of the saprolegnian oomycete clade (Fig. 4), and focused on the peronosporalean clade, which includes the well-established *Phytophthora* and *Pythium* genera, as well as the more basal *Albugo* spp. (*McCarthy & Fitzpatrick, 2017*). The tree topology was very consistent with previously published oomycete phylogenies (*Beakes, Glockling & Sekimoto, 2012*; *Lara & Belbahri, 2011*; *Spies et al., 2016*), and depicted several *Lagenidium* species within a monophyletic clade and as sister taxon to a cluster containing a strongly supported monophyletic grouping of *Phytophthora* spp. and a paraphyletic assemblage of *Pythium* lineages (Fig. 4). The branch leading to *Albugo* spp. remained basal to this *Phytophthora-Pythium-Lagenidium* cluster. Although all *Pythium* species appeared monophyletic, deeper nodes, indicative of relationships between various *Pythium* spp., were characterized by weak statistical support. Similarly, poor bootstrap support prevented the confirmation of a recently proposed *Lagenidium sensu stricto* classification that regrouped *L. giganteum*, *L. humanum* and *L. deciduum*, and was inferred from a six-gene phylogeny reconstructions that included *cox1* gene sequences (*Spies et al., 2016*). However, the present analysis confirmed the strongly supported, monophyletic association between *L. giganteum* and *L. humanum* (Fig. 4). All of the environmental sequences obtained from phytotelmata clustered within this *Lagenidium* clade, strongly validating the metagenomic approach, and the preliminary taxonomic identifications inferred from homology analyses. The environmental barcodes, independently from the amplification strategy and sequencing technology used to obtain them, segregated into two different groups: some sequences, including the most represented sequences generated using NGS technologies, appeared as sister taxa to *L. humanum* (99% bootstrap support), whereas another group of environmental sequences were strongly associated with the *L. giganteum* isolated from mosquito larvae (94% bootstrap support). Interestingly, no sequences appeared close to the *L. giganteum f. caninum* clade, or close to the more distant *L. deciduum* (Fig. 4), suggesting that, although the metabarcoding approach used in this study revealed a previously sub-sampled diversity within the genus *Lagenidium*, the sampling strategy may have biased the detection of *Lagenidium* spp. towards species that inhabit very specific ecological niches. The phylogenetic analyses clearly indicated that oomycetes such as *L. giganteum* and (possibly) *L. humanum* are present in phytotelmata, and that the metabarcoding approach described in this study

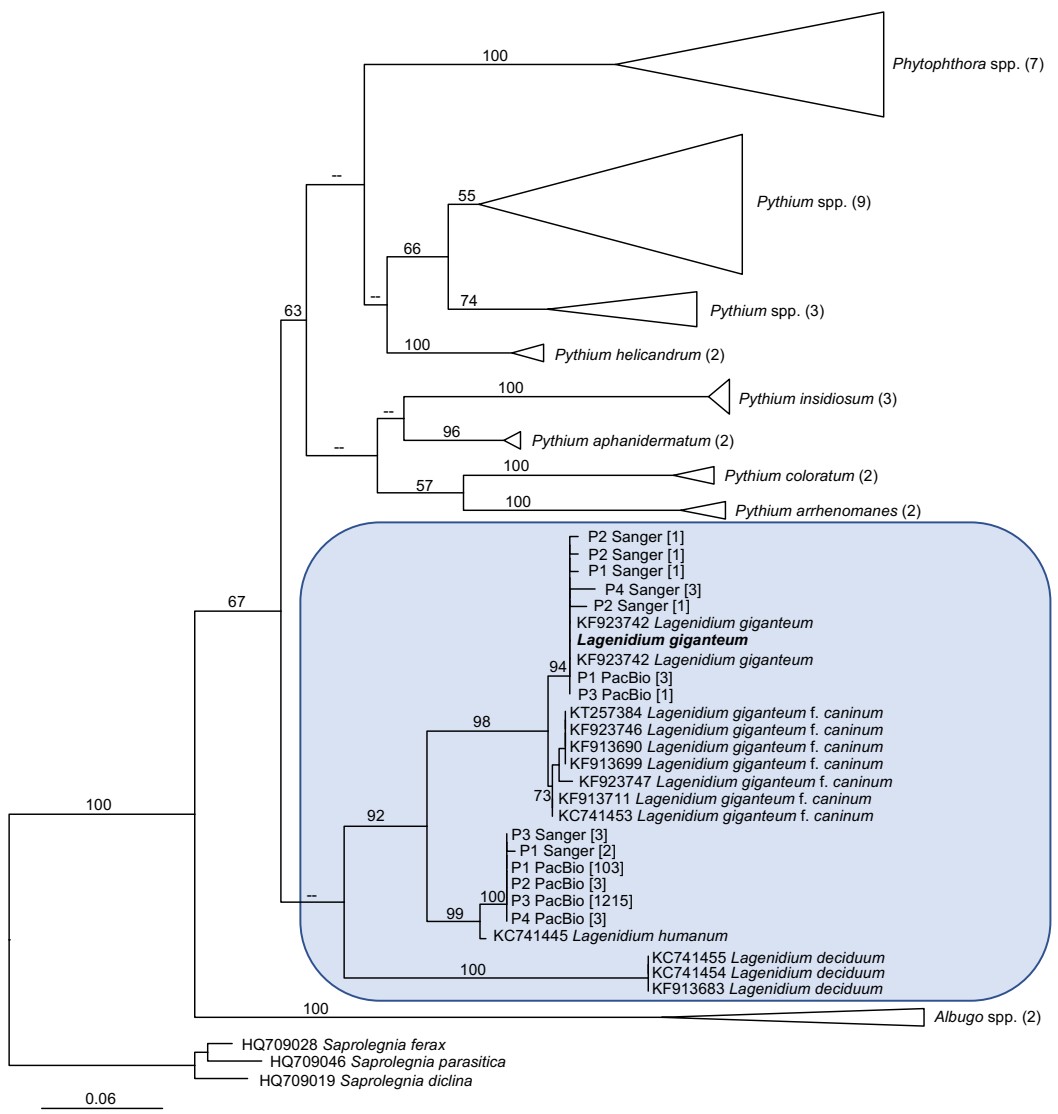

**Figure 4** **Maximum Likelihood (ML) phylogram inferred from oomycete *cox1* gene sequences, and incorporating environmental sequences generated using Sanger or PacBio sequencing strategies.** The origin of these environmental sequences is denoted by the codes P1–P4, corresponding to bromeliad phytotelmata 1 to 4, respectively. All other sequences were downloaded from public databases, except for the *Lagenidium giganteum* ARSEF 373 *cox1* DNA barcode (in bold) which was generated for this study. For environmental sequences, numbers in square brackets indicate the numbers of identical reads obtained throughout the metabarcoding analysis. For non-*Lagenidium* oomycete species, numbers in parentheses indicate the numbers of sequences used to generate the trees. Numbers at the nodes correspond to bootstrap values >50% (1,000 replicates), whereas less-supported nodes (<50%) are indicated with (–). The tree is rooted with *Saprolegnia* spp., and demonstrates that *Lagenidium* spp. barcodes were detected in all phytotelmata. All detected *Lagenidium* barcodes clustered within a strongly supported monophyletic clade that include *L. giganteum* and *L. humanum*.

provides a basis for the detection and isolation of novel *Lagenidium* strains independently of host-dependent baiting or occasional observations of infections.

## DISCUSSION

One of the major objectives of this study was to assess the presence of *Lagenidium giganteum* in phytotelmata. Two independent and complementary microbial detection strategies based on the amplification of *cox1* DNA barcodes were used and produced globally concordant outcomes that strongly suggested that *L. giganteum* can be detected in small aquatic environments such as phytotelmata, indicating opportunities for close associations not only with invertebrate hosts, but also with plant tissues. The use of a nested PCR strategy that integrated newly designed *Lagenidium*-specific primers generated a majority of sequences that clustered with the previously published *L. giganteum* cox1 gene fragments (Fig. 4), while high-throughput sequencing using a PacBio platform also produced *cox1* sequences consistent with the presence of *L. giganteum*. Overall, *L. giganteum* DNA barcodes were detected in all 4 sampled phytotelmata (Fig. 4). Furthermore, the two strategies were highly similar in highlighting the presence of potential additional *Lagenidium* species that appeared closer related to *L. humanum*. A single DNA barcode corresponding to a potentially novel *Lagenidium* phylotype was especially prevalent in the high throughput dataset, but was also detected as the only *Lagenidium* sequences in the P3 phytotelma by the alternate, nested-PCR-based protocol. Finally, although the sampling size of randomly-selected cloned *cox1* fragments sequenced through Sanger technologies remained modest, both detection methods were remarkable in failing to generate any sequences that have been associated with *Lagenidium* strains isolated from mammalian hosts. These multiple instances of concordance between methodologies contribute to strengthen the conclusion that specific *Lagenidium* phylotypes, including the entomopathogenic *L. giganteum*, are present in phytotelmata, and validate the use of the PacBio sequencing platforms (combined with *cox1* as DNA barcodes) as a potential strategy to assess oomycete community composition in environments of interest. Especially, the generation of identical Amplicon Sequence Variants (ASVs), with similarly high frequencies among *Lagenidium* spp. barcodes, in four independent plants serves to provide high levels of confidence in the quality of the datasets obtained using the SMRT strategy (*Callahan, McMurdie & Holmes, 2017*).

Comparisons between the two methodologies also revealed some discrepancies, highlighting the limitations of these detection techniques and the opportunity to use early oomycete metabarcoding analyses such as this study to devise more efficient protocols aimed at understanding oomycete communities in taxa-rich, complex substrates. Consistent with previous work (*Riit et al., 2016*), high throughput sequencing combined with broad range primers resulted in the amplification of non-target barcodes and, in the case of the P2 phytotelma, drastically decreased the sample size of oomycete reads used to assess the presence and relative frequencies of *Lagenidium* spp. (Fig. 3). Although the amplification of barcodes corresponding to microbial fauna representatives that are phylogenetically close to oomycetes (e.g., diatoms) appear difficult to eliminate, the generation of reads associated with animals or fungi suggests that the *cox1* primers, or the amplification conditions, used in this study may be refined to avoid non-target sequencing. Novel primer design sites in the *cox1* or other genes should be investigated to further the demonstrated potential of SMRT-based analyses, and favor the production of longer,

more oomycete-specific reads. In addition, combining PacBio sequencing with the use of the presented *Lagenidium*-specific primers and more constricted amplification conditions may offer a more thorough estimate of all *Lagenidium* phylotypes and their respective relative abundance, while limiting the production of sequences from other oomycetes and non-target organisms. A similar strategy was used previously for the plant pathogenic *Phytophthora*, and demonstrated that next generation sequencing technologies provide higher resolution compared to the traditional cloning/Sanger sequencing approaches, resulting in the detection of a higher number of phylotypes (*Prigigallo et al., 2016*). However, strategies based on genus specific primers do not offer the opportunity to globally assess oomycete communities. Approaches that combine both genus-specific and taxonomically broader primers are likely necessary to thoroughly appreciate the relative abundance of oomycetes such as *Lagenidium* spp. in plant microbiomes. Based on this study, the impact on *Lagenidium* spp. on potential invertebrate hosts within phytotelmata remains unclear, as they mostly appeared as low frequency members within oomycete communities, especially relative to *Pythium* (Fig. 3). This observation is consistent with previous metabarcoding analyses of soil oomycetes that demonstrated that Pythiales vastly outnumbered Lageniales (*Riit et al., 2016*). However, the read distribution obtained from P3 indicates that *Lagenidium* spp. relative frequency may rise under specific (and yet-to-be determined) circumstances, possibly associated with the presence of hosts, or other factors (Fig. 3). Within the genus *Lagenidium*, the relative abundance of multiple distinct phylotypes also remains unresolved: the *Lagenidium*-specific primers produces a majority of sequences that clustered with the *L. giganteum* OTUs (58% vs. 42% clustering with the *L. humanum* OTUs), but this observation was not supported by the PacBio sequencing data, which clearly identified *L. humanum* OTUs as the most abundant phylotype, with *L. giganteum* barcodes appearing only marginally (<1%). It remains unclear if the phylotype distribution obtained through high-throughput sequencing is an accurate representation of the *Lagenidium* spp. community within phytotelmata, or if it only reflects technical artefacts such as primer bias towards particular *cox1* barcodes. As mentioned above, these discrepancies offer the possibility to delineate more clearly-defined protocols for oomycete metagenomics.

Beyond the technical aspects, the presented study globally supports the hypothesis that *Lagenidium* spp. are present in phytotelmata and therefore provides novel insights on the ecological niches occupied by these poorly-known oomycetes. Investigating potential relationships with plant tissues within phytotelmata may reconcile the transcriptomics data that have blurred the distinction between plant vs animal pathogens (*Quiroz Velasquez et al., 2014*). The detection of *Lagenidium* spp. close to plant tissues also provides contextual support for the hypothesis that these oomycetes evolved from plant pathogens, and sheds light on a recurrent evolutionary pathway (shift from plant pathogenicity to entomopathogenicity) that has been observed independently in multiple, phylogenetically unrelated entomopathogens (*St Leger, Wang & Fang, 2011*; *Shen et al., 2019*). These observations can also be extended to *Py. insidiosum*, which appeared to have shifted from plant pathogenic ancestors and acquired the ability to cause infections in humans and other mammals (*Rujirawat et al., 2018*). The increasing interest in oomycetes

as animal pathogens, and the emerging diversity of oomycete hosts, place a previously unexpected emphasis on developing oomycetes as models for the study of evolution of pathogenic abilities and host selection.

Finally, the data generated in this study also highlights the value of culture-independent technologies to appreciate previously-unsampled oomycete diversity within the genus *Lagenidium*, and the potential of bromeliad phytotelmata as a source of novel mosquito biocontrol agents. The consistent generation of novel, similar oomycete ASVs in four independent plants suggests that a yet-to-be characterized *Lagenidium* phylotype may be isolated from phytotelmata, and since it inhabits demonstrated mosquito breeding sites (*Wilke et al., 2018*), may exhibit potential as vector biocontrol agent. Phylogenetic analyses revealed that this phylotype is more distant from the *L. giganteum* strains responsible for mammal infections, and therefore may prove to present less safety concerns than the *L. giganteum* isolates that were originally developed as commercial products, and currently abandoned (*Vilela et al., 2019*). The phylogenetic affinities exhibited by this potential new *Lagenidium* phylotype also offer the intriguing opportunity to investigate the potential of *L. humanum* as an invertebrate pathogen, and biocontrol agent. Despite its species name, *L. humanum* has never been reported as a human (or vertebrate) pathogen, but was originally and serendipitously isolated from soil samples using dead human skin pieces as baits (*Karling, 1947*). Its pathogenic abilities remain unknown and, because of the especially modest publication record focused on this species, it is also unclear if the material available from the ATCC (*Specker, 1991*) corresponds to the original isolate that was thoroughly described and illustrated in 1947 (*Karling, 1947*). Efforts to axenically isolate the major *Lagenidium* phylotype identified in phytotelmata, develop comparative analyses with *L. giganteum* and *L. humanum* strains maintained in culture collections, and evaluate the respective impact of these *Lagenidium* spp. on vector mosquitoes have been initiated.

## CONCLUSIONS

The phylogenetic reconstructions presented in this study were performed primarily to validate the metabarcoding analyses aimed at detecting *Lagenidium giganteum* in phytotelmata. A significant fraction of the DNA barcodes obtained through two independent methods corresponded to *Lagenidium* genes and clustered within a strongly supported, monophyletic clade that included both *L. giganteum* and *L. humanum*. Therefore, *Lagenidium* spp. are members of phytotelmata microbiomes. The development of such validated detection methods may not only be used to assess the prevalence and abundance of *Lagenidium* in relation to invertebrate host presence, but also serves as a basis to investigate potential relationships between *Lagenidium* phylotypes and their plant "hosts" (especially when invertebrate hosts, and water, are not present), and estimate the role of plant pathogenic-like oomycete effectors during these interactions. Finally, the metabarcoding analyses presented in this study revealed phytotelmata as promising sources for the identification of novel *Lagenidium* strains and/or species with potential as biocontrol agents against vector mosquitoes.

## ACKNOWLEDGEMENTS

Support for Next Generation Sequencing technologies was provided by Pacific Biosciences and the University of Florida Interdisciplinary Center for Biotechnology Research (ICBR).

### Funding

Funding for this project was provided through grants from the US Department of Agriculture (Agriculture and Food Research Initiative 2011-68004-30104) and a US Department of Education (Minority Science and Engineer Improvement Program P120A140012). The funders had no role in study design, data collection and analysis, decision to publish, or preparation of the manuscript.

### Grant Disclosures

The following grant information was disclosed by the authors:
US Department of Agriculture (Agriculture and Food Research Initiative 2011-68004-30104).
US Department of Education (Minority Science and Engineer Improvement Program P120A140012).

### Competing Interests

The authors declare there are no competing interests.

### Author Contributions

- Paula Leoro-Garzon, Andrew J. Gonedes and Isabel E. Olivera performed the experiments, analyzed the data, authored or reviewed drafts of the paper, approved the final draft.
- Aurélien Tartar conceived and designed the experiments, analyzed the data, contributed reagents/materials/analysis tools, prepared figures and/or tables, authored or reviewed drafts of the paper, approved the final draft.

### Data Availability

The *L. giganteum* cox1 sequence is available in the GenBank, EMBL, and DDBJ databases: MN099105.

The plant barcodes generated in this study are available at MN099106–MN099113, oomycete cox1 barcodes produced using the gene cloning/Sanger sequencing strategies are available at: MN099114–MN099125.

The sequences reads corresponding to cox1 barcodes generated using high-throughput sequencing (PacBio) technologies are available at NCBI Sequence Read Archive (SAR): SRX6359420, SRX6359421, SRX6359422, SRX6359423, (plants P1–P4, respectively) as part of Bioproject PRJNA550619.

## Supplemental Information

Supplemental information for this article can be found online at http://dx.doi.org/10.7717/peerj.7903#supplemental-information.

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
