# Peer review of "Oomycete metabarcoding reveals the presence of Lagenidium spp. in phytotelmata"

_PeerJ, doi:10.7717/peerj.7903_

## Round 0.1 · original submission · Minor Revisions

Three independent reviewers have finished their evaluation of your article and all of them have suggested minor changes.. Please, adress the changes suggested by the three reviewers.

Reviewer 1 ·

Basic reporting

No comment

Experimental design

The sequencing data certainly provides compelling evidence that Lagenidium spp are present in phytotelma, using two molecular methods to verify each other, and construction phylogenetic tree to complement sequencing. However, why was no attempt made to also culture any of the Lagenidium spp after sequences had been identified. Is it difficult to culture?

Validity of the findings

Could more information be provided on the relative abundance of all oomycete genera, not just Lagenidum (e.g. line 359). The read percent for Lagenidium was actually quite low for particularly P2 and P4. May be of interest to include read percent of the other oomycete genera - was the bulk of all the rest of the reads Pythium spp, and at what percent?

Line 464 - 467: Complementary approaches such as the ones presented.... Unclear what this means. This sentence implies that the role of oomycetes in phytotelma environment was evaluated using metabarcoding, but it was not. Next sentence goes on to state this. Suggest deleting or revising this sentence to indicate that environmental role was not ascertained in your study.

Line 476 - 479: It is not clear how this data is related to Fig 4, and also appears to be presenting new information in this discussion that wasn't previously shown in the results, i.e. differences in read percent of giganteum vs humanum in Sanger vs PacBio sequencing. Suggest including this information in the results first, rather than the discussion, and showing how the percentages were obtained.

Line 494 - 507: Although an interesting evolutionary discussion point, this section is not related to the original research question or to supporting results. Suggest shortening this section, or deleting it entirely.

Sequences have been deposited into GenBank, but at the time of this review, they are not yet available publically when searching the database.

Additional comments

Well written, easy to follow manuscript that describes previously unknown ecology of Lagenidium spp.

Reviewer 2 ·

Basic reporting

This manuscript presents research on oomycete communities in particular Lagenidium spp. in phytotelmata. The study provides new insights on the presence of Lagenidium species and their relations with other Lagenidium species studies earlier and is worthy of publication. The paper is well written and I enjoyed reading it.

Experimental design

The MS combines Sanger and amplicon sequencing technique targeting cox1 DNA barcodes. The materials and methods are appropriate, the data analysis and presentation are adequate, and relevant discussion.

Validity of the findings

The paper is a step forward in understanding oomycetes communities in plant microenvironment with focus on Lagenidium, its possible ecological significance, and overall covers an important contribution to the literature.

Additional comments

However, there are few concerns need to be addressed before publication. First one, the transcriptomics and genomic information related to Lagenidium and links to the pathogenicity evolution highlight the importance of this study, however, is over-used throughout the manuscript diverting the focus of the MS. The results shown in MS are solely based on DNA barcode and detection via sanger or amplicon sequencing. I suggest reducing the information on genomics, transcriptomics and pathogenicity pathways throughout the MS, especially in the discussion section (Fx line 487-510). Second, the methods section mentions about nested PCR, which needs clarification. Please explain two sets of primers, overlapping region, amplicon size, thermal cycle, and the main purpose behind it. Also, giving a name to the newly designed primer will help readers.
Another aspect is in the result section is that it is too descriptive. I suggest that you improve this section by reducing it, for instance, part of it describes methods, consider them to move in the materials and methods section. For example, line 279-results title says ‘Environmental detection of Lagenium spp’ but the actual result section starts from line 300. Similarly, PacBio sequencing (line 322), this section is difficult to follow with lots of details which could be easily removed or move to a table (fx line 323-329).

Some specific comments
Consider adding a line describing Phytotelmata in the introduction.
Line 296-967, presence of band doesn’t mean that it is always oomycetes based on your amplicon results (sample P2)
Line 168 consider adding ‘(NSU)’ after university
Line 365 remove the dot after sample

Reviewer 3 ·

Basic reporting

Use of English: I don't consider myself qualified to do a thorough check, but I haven't detected any serious issues.

The background provided is enough and the literature is relevant but quite out of date, presenting only 7 references among 51 published in the last 3 years. There are more up-to-date works published based on metabarcoding approaches for oomycete identification, as the followings:
Sapkota, R. & Nicolaisen, M. Cropping history shapes fungal, oomycete and nematode communities in arable soils and affects cavity spot in carrot. Agric. Ecosyst. Environ. 257, 120–131 (2018).
Abdelfattah, A., Malacrinò, A., Wisniewski, M., Cacciola, S. O. & Schena, L. Metabarcoding: A powerful tool to investigate microbial communities and shape future plant protection strategies. Biol. Control 120, 1–10 (2018).
Ruiz Gómez, F.J., Navarro-Cerrillo, R.M., Pérez-de-Luque, A., Oβwald, W., Vannini, A., Morales-Rodríguez, C. Assessment of functional and structural changes of soil fungal and oomycete communities in holm oak declined dehesas through metabarcoding analysis. Sci. Rep. 9, 5315 (2019)

Raw data (FASTA files and abundance matrices) are not supplied.

Experimental design

The research is original but the experimental design does not provide enough evidence to support some of the considerations the authors stated in the discussion. The research question is quite methodological and the number of samples and the chosen controls are not enough to provide evidence of the association of >Lageindium spp. and the plants, nor even the phytotelmata. Authors analyze water DNA, and they declare there is a significant amount of invertebrate and biological debris in the water. The results do not support the statement that Lagenidium spp is able to colonize phytotelmata, nor the plant tissues.

Also, there is another issue along with the text. Authors confounded barcode with the amplified sequence on several occasions. A barcode in NGS is the short specific sequence added to a primer in order to allow multiplexing and ulterior demultiplexing of DNA libraries.

Validity of the findings

The work is whort to be published, but some conclusions stated are not supported by the results and in my opinion, the discussion should be modified.

The analysis of the abundance matrix is not carried out, nor the biodiversity indices calculus for the oomycete community.

Statistical results for the phylogenetic analysis are not provided.

Annotated reviews are not available for download in order to protect the identity of reviewers who chose to remain anonymous.

---

## Round 0.2 · accepted · Accept

Most of the comments of the reviewers have been addressed correctly.